# Therapeutic Efficacy of Variable Biological Effectiveness of Proton Therapy in U-CH2 and MUG-Chor1 Human Chordoma Cell Death

**DOI:** 10.3390/cancers13236115

**Published:** 2021-12-04

**Authors:** Prerna Singh, John Eley, Nayab Mahmood, Binny Bhandary, Tijana Dukic, Kevin J. Tu, Jerimy Polf, Narottam Lamichhane, Javed Mahmood, Zeljko Vujaskovic, Hem D. Shukla

**Affiliations:** 1Division of Translational Radiation Sciences (DTRS), Department of Radiation Oncology, University of Maryland School of Medicine, Baltimore, MD 21201, USA; prerna.singh@umaryland.edu (P.S.); bbhandary@som.umaryland.edu (B.B.); tdukic@som.umaryland.edu (T.D.); jpolf@umm.edu (J.P.); narulamichhane@umm.edu (N.L.); javed.mahmood@bms.com (J.M.); zvujaskovic@som.umaryland.edu (Z.V.); 2Department of Radiation Oncology, Vanderbilt University School of Medicine, Nashville, TN 37232, USA; john.g.eley@vanderbilt.edu; 3College of Information Science, University of Maryland, College Park, MD 20742, USA; nayabsm217@gmail.com; 4Department of Cell Biology and Molecular Genetics, University of Maryland, College Park, MD 20742, USA; ktu@umd.edu

**Keywords:** chordoma, proton beam radiation, Middle of the Spread-Out Bragg Peak (M-SOBP), End of the Spread-Out Bragg Peak (E-SOBP), radiobiological effectiveness, linear energy transfer, radioresistance

## Abstract

**Simple Summary:**

Chordoma is a rare, slow-growing cancer of the spinal cord. Photon radiation therapy and surgery are the standard of care for chordoma. Proton radiation therapy has become an increasingly common treatment in comparison to photon radiation therapy due to the ability to reduce off-target radiation dose. However, there is still an increased risk of toxicity to the surrounding critical structures that lead to poor treatment outcomes. Moreover, the biologic effectiveness of protons to sterilize chordoma cells remains uncertain and likely varies according to the proton energy spectrum throughout the proton field. We aim to investigate the tumoricidal properties of proton radiation therapy at the middle and end of the proton radiation field and elucidate variations in the relative biological effectiveness for chordoma cells. Our study helps quantify the therapeutic value of treating chordoma near the end of the proton field, where linear energy transfer is relatively high.

**Abstract:**

Background: Chordoma is a cancer of spinal cord, skull base, and sacral area. Currently, the standard of care to treat chordoma is resection followed by radiation therapy. Since, chordoma is present in the spinal cord and these are very sensitive structures and often complete removal by surgery is not possible. As a result, chordoma has a high chance of recurrence and developing resistance to radiation therapy. In addition, treatment of chordoma by conventional radiation therapy can also damage normal tissues surrounding chordoma. Thus, current therapeutic options to treat chordoma are insufficient and novel therapies are desperately needed to treat locally advanced and metastatic chordoma. (2) Methods: In the present investigation, human chordoma cell lines of sacral origin MUG-Chor1 and U-CH2 were cultured and irradiated with Proton Beam Radiation using the clinical superconducting cyclotron and pencil-beam (active) scanning at Middle and End of the Spread-Out Bragg Peak (SOBP). Proton radiation was given at the following doses: Mug-Chor1 at 0, 1, 2, 4, and 8 Gy and U-CH2 at 0, 4, 8, 12, and 16 Gy. These doses were selected based on a pilot study in our lab and attempted to produce approximate survival fractions in the range of 1, 0.9, 0.5, 0.1, and 0.01, respectively, chosen for linear quadratic model fitting of the dose response. (3) Results: In this study, we investigated relative biological effectiveness (RBE) of proton radiation at the end of Spread Out Bragg Peak assuming that the reference radiation is a proton radiation in the middle of the SOBP. We observed differences in the survival of both Human chordoma cell lines, U-CH2 and MUG-Chor1. The data showed that there was a significantly higher cell death at the end of the Bragg peak as compared to middle of the Bragg peak. Based on the linear quadratic (LQ) fit for cell survival we calculated the RBE between M-SOBP and E-SOBP at 95% CI level and it was observed that RBE was higher than 1 at E-SOBP and caused significantly higher cell killing. Proton field at E-SOBP caused complex DNA damage in comparison to M-EOBP and the genes such as DNA topoisomerase 1, *GTSE1*, *RAD51B* were downregulated in E-SOBP treated cells. Thus, we conclude that there seems to be substantial variation in RBE (1.3–1.7) at the E-SOBP compared with the M-SOBP.

## 1. Introduction

Radiation is a common treatment modality for cancer [1], and photon-based radiation therapy is conventionally used, however, due to the limited mode of delivery via multiple X-ray beams, there is an increased risk of toxicity to surrounding normal tissue [2]. Chordoma is a rare slow-growing cancer of the spinal cord which occurs more commonly on the sacral, followed by cranial and thoracic-lumber regions [3]. Hence, due to the critical position of the chordoma tumor and the limitation of photon radiation, proton beam radiation therapy (PBRT) has become standard of care [4]. PBRT has become an increasingly available treatment modality for several tumors, owing to its dose characteristic, known as Bragg peak, which occurs before the range where protons come to rest [5,6]. The Bragg peak is observed when the absorbed dose increases very gradually with greater depth, suddenly rising to a peak just before the protons come to rest (Figure 1) [7]. The Bragg peak is depicted through a Bragg curve which portrays the energy loss of ionizing radiation as it traverses through matter. However, the high dose region of the Bragg peak itself only covers a narrow range of depth near the end of the beam range. In order to widen the treatment depth range, proton beams with multiple initial starting energies are stacked into what is known as the Spread-out-Bragg Peak (SOBP). Protons are charged particles that gradually lose their speed as they penetrate human tissue and deposit energy. Moreover, proton therapy allows the targeting of tumors inside the body via precise localization of the radiation dosage which in turn helps spare the patient’s healthy cells, offering a much less invasive alternative to treat cancer [8].

Ionizing radiation like proton and photons cause cell death either by directly and/or indirectly damaging the nuclear DNA, which leads to impairment of the DNA repair pathway. The lethality of a charged particle like protons, on a cell by nuclear DNA damage depends greatly on the linear energy transfer (LET) of the charged particle and the dependence is usually expressed by the relative biological effectiveness (RBE) [9]. RBE is defined as the ratio of photon/proton RT doses to achieve the same level of effect when comparing two radiation modalities, for example, 250 kVp X-ray photon radiation as the reference radiation and proton radiation as the experimental radiation. Numerous clinical treatment centers assume a fixed proton RBE with a value of 1.1 (^60^Co gamma rays reference) whereas the RBE value of low-LET X-rays is 1.0, and it is known that higher relative biological effect results at the End of Proton Spread-out-Bragg Peak (E-SOBP) [10,11,12]. LET is measured in keV/μm and is the average radiation energy deposited per unit length traveled along the ionizing particle and largely determines the biological response of a cell to irradiation. Photons are considered lower LET at 0.2–3 keV/μm whereas heavy particles such as protons are at higher LET at 0.5–30 keV/μm; higher LET is known to cause higher biological direct damage than lower LET. Hence, the higher RBE measured in the E-SOBP regions is often attributed to higher LET of the proton beam in this region [13,14].

The precise estimation of biological effectiveness of protons as a function of energy has not been fully investigated. Moreover, the RBE data have been acquired for only a handful of cell lines, tissues, and endpoints. The energy of proton beam at the middle and end of Bragg peak and its energy distribution, mainly depends on the target depth, considerably affects the track structure of the protons that is crucial for DNA damage [15,16].

Nevertheless, chordoma is a highly radioresistant cancer, requiring doses > 70 Gy. Despite the highly conformal properties of proton radiation, there is still a minimal number of protons deposited to the surrounding tissue, which becomes profound with higher dosage. In the present investigation we compared chordoma cell killing in the middle and end of SOBP and their relative biological effect. Further, we also compared gene expression profile of U-CH2 and MUG-Chor1 to identify differentially expressed genes at the middle and end of spread-out Bragg peak. In order to elucidate the relative biological effectiveness between M-SOBP and E-SOBP, we applied LQM to analyze survival fraction (SF) and derived parameters alpha and beta of LQM from an in vitro SF cells experiment.

## 2. Materials and Methods

### 2.1. Cell Culture

Human chordoma cell lines of sacral origin MUG-Chor1 and U-CH2 were obtained from the American Type Culture Collection (Manassas, VA, USA) and cultured at 37 °C in Iscove’s modified Dulbecco’s medium (IMDM/RPMI (4:1; Gibco; Thermo Fisher Scientific, Inc., Waltham, MA, USA) supplemented with 10% Fetal Bovine Serum (FBS; Invitrogen, Thermo Fisher Scientific, Inc.) and 1X Antibiotic-Antimycotic. Cells were cultured in 50 µg/mL collagen coated T-75 flasks (Corning, Bedford, MA, USA) with a doubling time of 5–7 days. The tissue culture plates were treated with 50 µg/mL collagen with autoclaved deionized water at room temperature for 1 h and then the plates were washed with 1 × PBS.

### 2.2. Proton Beam Radiation Treatment

Chordoma cells were irradiated using the clinical superconducting cyclotron and pencil-beam (active) scanning gantry (ProBeam, Varian Medical Systems, Palo Alto, Santa Clara, CA, USA) at the Maryland Proton Treatment Center (Baltimore, MD, USA). For each exposure, a treatment plan consisting of a rectangular treatment field with a 9 × 14 cm^2^ irradiation field with a 10 cm SOBP width was developed using the Eclipse treatment planning system (V15.6, Varian Medical Systems). To deliver the SOBP the planes treatment field delivered proton beams with 22 different initial energies ranging from 148 to 80 MeV with the dose delivered by each beam energy weighted so that a uniform dose was delivered across the entire width of the SOBP (Figure 2B). The proton gantry was rotated to deliver the beam from beneath the plates, and tissue-equivalent plastic slabs were placed beneath the 6-well plates (Figure 2A) to position the cell-culture plates at either M-SOBP (9.7 cm water equivalent depth in plastic) or E-SOBP (14.7 cm water equivalent depth in plastic). Proton radiation was given at physical doses: Mug-Chor1 at 0, 1, 2, 4, and 8 Gy and U-CH2 at 0, 4, 8, 12, and 16 Gy. These doses were selected based on a pilot study in our lab and attempted to produce approximate survival fractions in the range of 1, 0.9, 0.5, 0.1, and 0.01, respectively, chosen for linear quadratic model fitting of the dose response. All proton dose exposure conditions were verified using a calibrated parallel plate ionization chamber (PPC05, IBA, Herndon, VA, USA) placed at the location of the 6-well plates, following the IAEA TRS-398 protocol for proton dosimetry but in tissue plastic phantom. In addition, a measurement of the SOBP used in the experiment can be seen in Figure 2, taken via a multi-layer ionization chamber (Giraffe, IBA, Ottignies-Louvain-la-Neuve, Belgium). The experimental groups were no treatment, proton radiation treatment at the M-SOBP, and radiation at E-SOBP, where number of replicate samples (*n*) was 18 (*n* = 18) for each delivered dose.

### 2.3. Colony Forming Assay

Cells were seeded in collagen coated 6-well tissue culture plates at densities 3000 cells/well and incubated at 37 °C and 5% CO_2_ (Airgas, Linthicum Heights, MD, USA) for 24 h prior to radiation treatment. After treatment, plates were incubated for 14 days. Colonies were fixed and stained with 0.05% *w/v* crystal Violet (Sigma Aldrich, St-Louis, MO, USA) in 37% formaldehyde. Colonies were counted using ProtoCOL3 colony counter (synbiosis, Cambridge, UK).

### 2.4. Proton Radiation Treatment

Based on our pilot scale study, UCH2 and Mug-Chor1 human chordoma cells were treated with 4 Gy of proton beam radiation at MOBP and EOBP. After treatment, cells were processed for RNA extraction and purification.

### 2.5. RNA Extraction and Purification

Total RNA was extracted using the Trizol reagent (Life Technologies, Rockville, MD, USA), according to the manufacturer’s instructions. Following homogenization, 1 mL of solution was transferred to a 1.5 mL Eppendorf tube and centrifuged at 12,000× *g* for 10 min at 4 °C to remove particulate material. The supernatant containing RNA was transferred to a new tube, mixed with 0.2 mL of chloroform (Sigma Aldrich, St-Louis, MO, USA), and centrifuged at 12,000× *g* for 15 min at 4 °C. The aqueous phase containing RNA was transferred into a new tube and RNA was precipitated by mixing 0.5 mL of isopropyl alcohol (Sigma Aldrich, St-Louis, MO, USA) and precipitated by centrifuging the tube at 12,000× *g* for 10 min at 4 °C. The RNA pellet was washed briefly in 1 mL of 75% ethanol (Sigma Aldrich, St-Louis, MO, USA) and centrifuged at 7500× *g* for 5 min at 4 °C. Finally, the total RNA pellet was dissolved in nuclease-water, and its purity was checked using Nanodrop (ThermoFisher Scientific, Watham, MA, USA).

### 2.6. Microarray Analysis

Microarray U133 Plus 2.0 processing and hybridization: Microarray U133 features 1,300,000 unique oligonucleotides, which include 47,000 transcripts, and represents roughly 39,000 human genes. Preparation of cRNA, hybridization, and scanning of microarrays were accomplished according to the Affymetrix protocol. In brief, 5 µg of total RNA was converted into double-stranded cDNA by reverse transcription. Biotin-labeled cRNA was generated by converting the cDNA sample using a BioArray High Yield RNA Transcript labeling kit (Enzo Life Sciences, Farmingdale, NY, USA). Labeled cRNA was hybridized to the Affymetrix U133 Plus 2.0 GeneChip (Affymetrix Inc., Santa Clara, CA, USA) while rotating at 60 rpm for 16 h at 45 °C. After hybridization, the microarray was washed using the Affymetrix Fluidics Station (Affymetrix Inc., Santa Clara, CA, USA) according to the manufacturer’s protocol. The chips were scanned in an Affymetrix 3000 7G scanner (Affymetrix Inc., Santa Clara, CA, USA). This array contains 46,228 probes comprising 7815 probe sets. The Flash Tag Biotin HSR kit (Genisphere, (Affymetrix Inc., Santa Clara, CA, USA) was used according to manufacturer’s protocol to label the miRNA. In brief, the process starts with total RNA containing low molecular weight RNA and the procedure begins with a brief tailing reaction followed by ligation of the biotinylated signal molecule to the target RNA sample. The labeled sample was then hybridized to the Affymetrix miRNA GeneChip while rotating at 60 rpm for 16 h at 48 °C. After hybridization, the microarray was washed using the Affymetrix Fluidics Station according to the manufacturer’s protocol. The chips were scanned in an Affymetrix 3000 7G scanner. Raw data generated as CEL files by the Affymetrix Expression Console (Affymetrix Inc., Santa Clara, CA, USA) were imported for normalization and analysis into Partek Genomics Suite v7.0 (Partek Inc., St Louis, MO, USA). Data were extracted using the Affymetrix HG-U133_Plus_2 Human na36 transcript annotation (Affymetrix Inc., Santa Clara, CA, USA), log2 transformed, and quantile normalized with the RMA (Robust Multi-Array Average) algorithm. This yielded data for 44.6 K transcripts, 21,346 unique NCBI Entrez genes, whose annotation was subsequently updated to current HGNC/NCBI nomenclature. After data quality control the samples’ gene expressions were independently compared using a two tailed one or two-way ANOVA between their biological classes, which provided relative levels of gene expression and statistical significance, as fold change and *p*-value, respectively. Genes whose log2 fold changes were greater than 2SD from the mean, essentially unchanged, were deemed to be differentially expressed. These differentially expressed genes then underwent Canonical Pathway analysis using the QIAGEN Ingenuity Pathway Analysis platform (IPA, QIAGEN, Redwood City, CA, USA) (www.ingenuity.com; accessed on 3 April 2020) to evaluate the biological pathways likely perturbed by these genes’ expression. The bioinformatics analysis was performed at Johns Hopkins Transcriptomics and Deep Sequencing Core Facility.

### 2.7. Statistical Analysis

Prior to the analysis, each survival fraction datum was log-transformed to ensure a good model fit at the higher dose levels. The paired-sample two-tailed Student’s *t*-test on Excel (α = 0.05, β = 0.2) was performed to analyze and compare distribution of differences in means for end and middle of spread-out Bragg peak. A statistically significant difference was considered when *p* ≤ 0.05 and highly significant *p* ≤ 0.01. The linear quadratic (LQ) models are essential in radiation biology and cell survival is represented by the formula:(1)$$\mathrm{SF}={\;e}^{-\left(\alpha D+{\;\beta D}^{2}\right)}$$
where D is the radiation dose and SF is the surviving fraction. The α and β are parameters of the linear and quadratic component, respectively, of the LQ model that determine radiation sensitivity. The biological rationale of LQ model is based on radiation-induced DNA double-stranded breaks that result in cell death [17]. Using the fitted cell survival parameters α and β for the E-SOBP and M-SOBP conditions, the relative biological effectiveness was calculated: (2)$$\mathrm{RBE}(D_{E})=\frac{-\beta_{M}\pm \sqrt{\beta_{M}^{2}+4\alpha_{M}\left(\alpha_{E}D_{E}^{2}+\beta_{E}D_{E}\right)}}{2\alpha_{M}D_{E}}$$
where *D_E_* is the physical radiation dose at the E-SOBP.

## 3. Results

To treat the cells at the middle or end of SOBP we selected and optimized rectangular irradiation field with a 10 cm SOBP width and rectangular geometry matching the rectangular border of the 6-well plates (Figure 2A), using the Eclipse treatment planning system (Varian Medical Systems), to uniformly irradiate the entire 6-well plate with a given proton dose. The proton gantry was rotated to deliver the beam from beneath the plates, and tissue-equivalent plastic slabs were placed beneath the plates (Figure 2A,B) to position the cell-culture plates at a radiologic depth corresponding to the middle of the SOBP.

### 3.1. E-SOBP Treatment Enhanced Significant Killing of Chordoma Cells Compared to M-SOBP Treatment

The clonogenic cell survival assay demonstrated enhanced cell killing of E-SOBP compared to M-SOBP in the Chordoma cell line model. The difference in survival fraction for both chordoma cell lines U-CH2 and MUG-Chor1 was statistically significant (*p*-value ≤ 0.05) for proton radiation doses at 4, 8, and 12 Gy, and 1, 2, and 4 Gy, respectively, between middle and end of SOBP (Figure 2C,D). However, the highest lethal dose 16 Gy for U-CH2 and 8 Gy for Mug-chor1 did not show any statistically significant killing because the dose was so high that the survival fraction at M-SOBP was < 1%. The data showed enhanced cell killing at the end of E-SOBP compared to M-SOBP. The data presented in Figure 2B demonstrated that UCH2 cells could survive at twice as high dose of proton radiation than MUG-Chor1 cells which could be due to strong DNA repair system and radiation resistance in U-CH2 cells. However, the decrease in chordoma cell survival at the E-SOBP was far more significant as compared to M-SOBP at lower doses. Thus, this suggests that there is a significantly higher cell killing at lower doses at E-SOBP than at M-SOBP.

### 3.2. Comparison of Relative Biological Effect in M-SOBP and E-SOBP

The relative biological effectiveness (RBE) of proton beam radiation varies in the whole range of Bragg peak and in the current investigation we compared RBE at the end of Bragg peak and middle of Bragg peak employing UCH-2 and MUG-Chor1 chordoma cell line models. For this work, we chose M-SOBP for reference condition to use for *D_ref_* for RBE calculation. While this condition deviates from much of the previous RBE work using 250 kVp X-ray’s reference, we chose to use it because firstly, it requires a minimal number of free variables in study, proton energy varies but not the irradiation machine. Secondly, it allows direct relative comparison between proton effectiveness at different locations along the path of the proton beam. Proton therapy is the treatment of choice for chordoma patients, and we made a direct comparison that could be used to see maximum killing effect and support the use of higher linear energy transfer (LET). The RBE was calculated for *D_E_* = 2 Gy at E-SOBP, since 2 Gy radiation is a clinically relevant dose. A linear quadratic (LQ) fit for cell survival was generated using gnuplot to calculate between M-SOBP and E-SOBP (Figure 3A,B and Table 1 and Table 2).

A linear quadratic (LQ) fit for cell survival was generated to calculate the RBE between M-SOBP and E-SOBP. Thus, the data delineated RBE was higher at E-SOBP as compared to M-SOBP, indicating more cellular damage than at M-SOBP. As a standard practice, 2 Gy radiation is commonly given in the clinic and at this 2 Gy level, the RBE at E-SOBP was also higher at 1.67 for U-CH2 and 1.32 for Mug-chor1. This suggests there is a significantly higher RBE at the E-SOBP than the M-SOBP in chordoma cell lines killing.

The data shows that there was greater cell killing at the end of SOBP compared to M-SOBP which was correlated with significantly higher RBE at the E-SOBP than the M-SOBP in chordoma cell lines at 95% survival. The RBE at E-SOBP is greater than 1 likely indicating more cellular damage and cell death than M-SOBP alone (Table 1). The 2 Gy radiation is commonly given in the clinic and according to 2 Gy the RBE at E-SOBP is also higher at 1.67 for U-CH2 and 1.32 for Mug-chor1 (Table 2).

### 3.3. Comparative Gene Expression Analysis of UCH2 Cells Treated at E-SOBP and M-SOBP of Proton Radiation

We also investigated gene expression pattern in UCH2 chordoma cells treated with proton radiation (PBRT) at middle and end of Bragg peak. Gene expression analysis was performed with GeneChip^®^ Human Genome U133 plus 2.0 arrays in M-SOBP and E-SOBP treated cells. The comparative gene expression pattern revealed significant changes in gene expression pattern (Figure 4). Notably, we found a high number of genes differentially expressed in E-SOBP as compared to M-SOBP, and a relatively small number of genes were commonly present in both. Notably, in E-SOBP cells, we observed upregulation *CASP8* molecular switch for apoptotic pathway, the anti-proliferative tumor regulator *CDKN1C*, and tumor suppressor and Ras Suppressor Protein 1, which is also a strong tumor suppressor gene in E-SOBP treated cells. In addition, we also identified downregulation of homologous recombination gene *RAD51B* in E-SOBP cells, which impacts cellular ability to repair lethal double strand breaks in chordoma cells.

The bioinformatic Ingenuity Pathway Analysis tool (Qiagen Inc.) also identified downregulation of OX40 L pathway in UCH2 treated at the end of Bragg peak (E-SOBP). This pathway is CD134 also known as TNF alpha super family which could be targeted using agonistic anti-OX40 antibodies to promote T cell activation and kill chordoma cells (Figure 5). This is the first report of the existence of a novel checkpoint pathway in chordoma cells.

## 4. Discussion

Proton beam radiation therapy is increasingly used in clinical settings worldwide compared to photons. The present study aimed to investigate variation in relative biological effectiveness between the middle and end of SOBP of proton radiation in chordoma cell killing. We used MUG-Chor1 and U-CH2 chordoma cell lines and treated with different doses of PBRT at the middle and end of SOBP and cell survival was determined employing clonogenic survival assay. We treated MUG-Chor1 cells with 0, 1, 2, 4, and 8 Gy and U-CH2 cells with 0, 4, 8, 12, and 16 Gy at M-SOBP and E-SOBP of proton radiation based on the dose optimized in our pilot study. The data demonstrated higher cell death of both MUG-Chor1 and U-CH2 cell lines at the E-SOBP as compared to M-SOBP. We further used this survival data to model a linear quadratic model to determine the α and β components and then compare the RBE of M-SOBP vs. E-SOBP. Variation in RBE is observed across the SOBP, with the RBE higher than the clinically used value of 1.1 for protons. Based on the clinically relevant dose of 2 Gy, we calculated RBE value of 1.67 at E-SOBP for UCH2, and 1.32 for Mug-Chor1 as relative to M-SOBP (Figure 3A,B and Table 1 and Table 2). Notably, we observed that RBE increased at the E-SOBP which causes severe irreversible cellular damage to the cancer cells as compared to M-SOBP [18]. This in vitro study corroborates that clinical optimization of variable RBE at the E-SOBP could be immensely important to optimize the proton field to maximize chordoma cell killing and minimize normal tissue damage [19,20]. The current proton therapy dose prescription for chordoma allows more precise targeting to treat chordoma. Notably, in contrast to photon-based therapy, proton therapy has offered relatively improved long-term local control and survival. However, in some recent clinical trials, proton therapy treatment has not been very effective and has caused recurrence in 25% of cases within three years. Thus, more studies are needed to draw important clinical conclusions.

Further, the implication for clinical dose prescription is that physical doses of proton therapy can be reduced in regions of the proton field having a higher RBE and that this reduction of physical dose could reduce damage to healthy normal tissues in the distal regions of a proton field. We expect this concept most likely to advance within the framework of intensity modulated proton therapy, where the intensity of doses can be readily adjusted throughout the irradiated region. Furthermore, to enable quantitative adjustment of physical doses to correspond to the regions of higher RBE, an RBE-modeling framework is needed within the treatment planning system. The framework is also needed to translate the in vitro RBE values into useful in vivo RBE values, accounting for the desired fractionation of treatment, the physical dose, and the LET spectrum, which vary throughout the patient. Such a framework is used clinically already for carbon-ion therapy [21,22] and might be adapted for proton therapy.

It has been noted that both cells exhibited differential sensitivity to proton radiation, suggesting that chordoma patients will respond differently to the same amount of proton dosage regardless of the position on SOBP. However, for both cell lines, the cellular damage is greater when E-SOBP is employed. The relationship between RBE and LET is not linear, RBE increases as LET increases until 100 KeV/μm and decreases with high LET afterwards due to cellular “overkill”. The present study strongly suggests that the higher cancer cell death at fall off (E_SOBP) of the Bragg peak could be due to high Linear Energy Transfer (LET) [13,18,23].

Furthermore, gene expression profiling analysis of chordoma cells in M-SOBP and E-SOBP was also examined employing UCH2 cells to unravel the molecular impact of increased RBE in causing more severe DNA damage to cancer cells which could be more challenging for the cancer cells to repair and that may lead to higher cell death [24,25]. The comparative gene expression data presented in Figure 4 demonstrated distinct pattern of gene expression in middle vs. end of SOBP. The heat map showed a significant differential gene expression pattern at the end of the Bragg peak when compared to middle of SOBP. We observed significant upregulation of the caspase 8 (*CASP8*) gene which is responsible for the apoptotic pathway and *CDKN1C* which is a tumor suppressor gene. We also observed downregulation of *RAD51B* at E-SOBP which is responsible for homologous recombination. We envisaged that E-SOBP might be causing severe DNA damage which could lead to irreparable DNA damage and cell death [26]. The Ingenuity Pathway Analysis of the gene expression data at both M-SOBP and E-SOBP have shown downregulation of OX40 ligand in OX40 pathway. Thus, CD134 could be a novel target to treat chordoma by blocking OX40 ligand using agonist OX40 antibodies. Thus, we conclude that this novel checkpoint could be more promising to treat chordoma patients as compared to anti-PDL-1 checkpoint therapy [27].

One limitation of this study was that we studied two distinct locations within the SOBP to represent, firstly, a control condition in the middle of the SOBP, where we expected the proton field would be most representative of the average clinical exposure condition, and, secondly, near the distal end of the high-dose region of the spread-out Bragg peak, where we expected to find the highest RBE values relevant for high-dose regions. A future systematic investigation of RBE for variable dose/LET conditions would benefit the future RBE modeling efforts needed to benchmark RBE-based treatment planning systems.

## 5. Conclusions

Thus, we conclude that relative biological effect was found to be higher at end of the Bragg peak which could be clinically more relevant than the middle of the Bragg peak. The gene expression data also showed downregulation of RAD51B gene at E-SOBP of proton treatment, which is unable to activate recombinational repair, and therefore causes more chordoma cell death. We also identified a downregulated OX40 ligand pathway (Figure 5), which is a novel checkpoint pathway in E-SOBP treated UCH2 cells. Thus, agonist anti-OX40 antibodies could be used to block OX40 receptor from binding its ligand and stop chordoma cell growth.

## Figures and Tables

**Figure 1 cancers-13-06115-f001:**
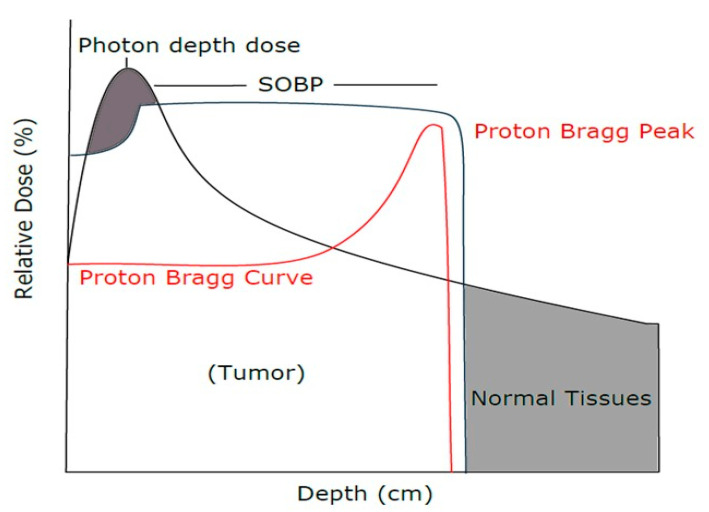
Comparison of photon and proton dose distribution in normal and tumor tissues.

**Figure 2 cancers-13-06115-f002:**
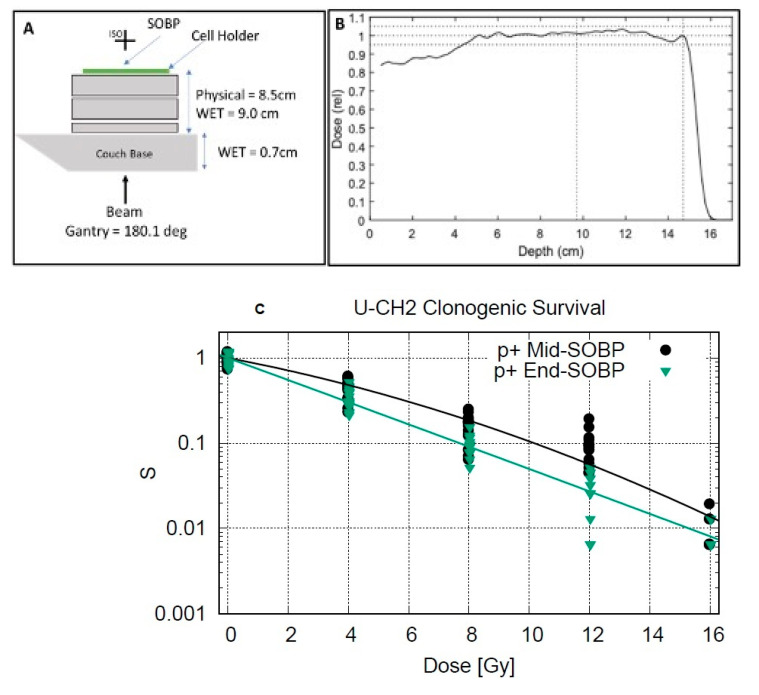
(**A**,**B**) Schematic diagram to show proton radiation treatment to cells. Cell survival of chordoma cell lines (**C**) U-CH2 and (**D**) MUG-Chor1 after proton radiation at the M-SOBP and E-SOBP.

**Figure 3 cancers-13-06115-f003:**
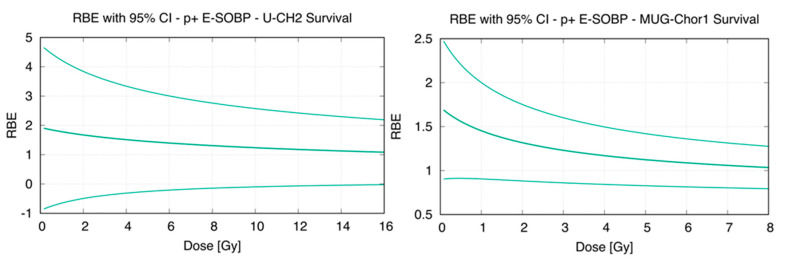
Relative Biological Effectiveness (RBE) of proton beam in the end of spread-out Bragg peak compared to that of the Middle of the Spread-Out Bragg Peak for UCH2 (**left**) and MUG-Chor1 (**right**) chordoma cell killing.

**Figure 4 cancers-13-06115-f004:**
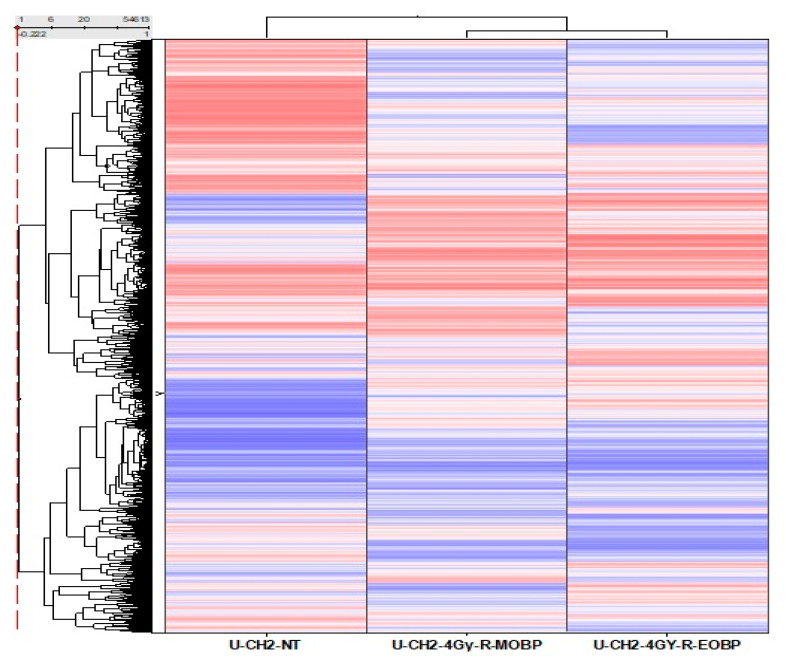
Gene expression profile of U-CH2 chordoma cells treated at middle (M-SOBP) and end of the Bragg (E-SOBP) peak of proton radiation.

**Figure 5 cancers-13-06115-f005:**
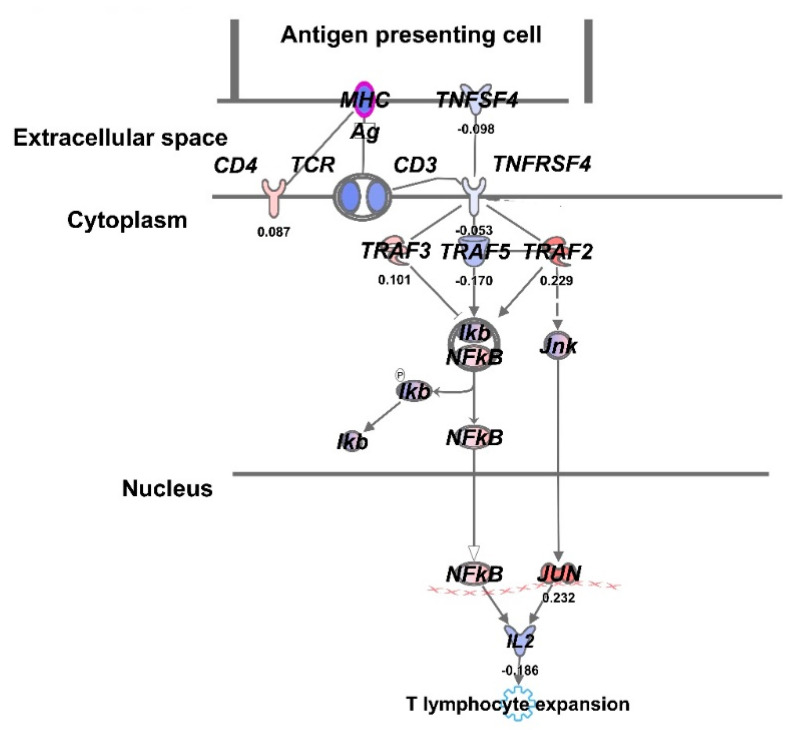
Identification of novel OX-40 check point signaling pathway in U-CH2 chordoma cells treated with end of Bragg peak proton radiation. Phosphorylation (**p**) of Ikb kinase tightly inhibits NFkB activation and its downstream adverse effects.

**Table 1 cancers-13-06115-t001:** UCH2 LQ fit and RBE (D_E_ = 2 Gy).

UCH2	α (Gy-1) (95% CI)	β (Gy-2) (95% CI)	RBE (D = 2 Gy) at 95% CI
M-SOBP	0.155 (0.114, 0.195)	0.007 (0.004, 0.010)	1
E-SOBP	0.298 (0.260, 0.337)	0.000 (−0.003, 0.003)	1.67 (−0.49–3.84)

**Table 2 cancers-13-06115-t002:** Mug-Chor1 LQ fit and RBE (D_E_ = 2 Gy).

MUG-Chor1	α (Gy-1) (95% CI)	β (Gy-2) (95% CI)	RBE (D = 2Gy) 95% CI
M-SOBP	0.243 (0.177, 0.310)	0.043 (0.033, 0.052)	1
E-SOBP	0.419 (0.352, 0.485)	0.025 (0.015, 0.034)	1.32 (0.88, 1.75)

## Data Availability

Data will be made available upon reasonable request to the corresponding author.

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
