# Peer review of "Therapeutic Efficacy of Variable Biological Effectiveness of Proton Therapy in U-CH2 and MUG-Chor1 Human Chordoma Cell Death"

_cancers, 2021, doi:10.3390/cancers13236115_

Round 1
Reviewer 1 Report
In this monocenter study, the authors prospectively analyzed human chordoma cell lines of sacral origin after irradiation with protons. The authors investigated relative biological effect of proton beam radiation at Middle (M-SOBP) and End (E-SOBP) of the Bragg Peak. This study observerd differences in survival of both chordoma cell lines and shown significantly higher cell death at the end of the Bragg Peak.
This topic is very attractive for the radio-oncological community because these results can improve the current dosage concepts
The manuscript is clearly structured and very well written. The tables are well formatted and of high quality. The quality of figures is good.
In my opinion, the manuscript may be suitable for publication provided that some issues are considered and reviewed:
- Please add to the discussion what impact the results may have on current dose prescriptions.
- Do the authors see a correlation between, for example, brain necrosis and overlap of radiation fields when irradiating skull base chordomas?
Author Response
Dear Sir,
Thanks for reviewing the manuscript and we have addressed all comments raised by you. We are sure this manuscript will attract wide scientific audiance.
Sincerely,
Hem D Shukla

Reviewer 2 Report
In a clinical practice of proton therapy treatment planning is accepted constant RBE =1.1 through the target volume. However it is well known that RBE can be essentially more than 1.1 at depth close to the end of the SOBP and especially at distal part of the SOBP. Increasing RBE at end of the SOBP is leading to an effective range increasing and possible overdosing of OAR close to the target volume . On the other hand increasing of RBE can be used for more efficient treatment of the radioresistant tumors.
On this point of view the paper is timely and in line with many other recent papers considering importance of variable RBE in proton therapy planning. Application of higher LET region of the SOBP for treatment of Chordoma is innovative however uncertainty in a range of protons should be considered and discussed by authors due to critical structures are closed to the target. It is not clear why authors are considering a single depth point for RBE study. RBE is changing fast with small depth increments downstream of this point that will be realised in higher GyEq dose that at this point. Demonstration at least at several depth points will be beneficial for suggested applications as a target is a not a single point. (see, for example , https://pubmed.ncbi.nlm.nih.gov/27084630/ and https://pubmed.ncbi.nlm.nih.gov/30468682/ ).
Quality of graphs to be improved to required standard for this journal. SF curves to be presented in a log scale as accepted in radiobiological practice. Some sentences were started but not completed.
More comments please see below.
Advantage of this work is in analysing of the gene expression data and demonstration of down regulation of RAD51B gene at the region of higher RBE, i.e. at E-SOBP .The same is possible to say about findings related to down regulated OX40 ligand pathway. Both mentioned findings are essential for chordoma treatment improvement. Statistical analysis is satisfactory.
Manuscript has many mistakes and some sentences were not clear worded . I would suggest rewriting the manuscript taking into account mentioned comments followed by editing by a native English speaker.
In summary, results are interesting and can be basis for a good and interesting paper but I can’t recommend this manuscript for publication in a current version for benefit of the authors.
Specific Comments.
Abstract
…. cultured irradiated with Proton Beam Radiation….change to …… cultured and irradiated with a proton beam
we investigated relative biological effect (RBE)…..abbreviation RBE is accepted as Radiobiological Efficiency . Change to ….we investigated RBE of the proton radiation field at the end of the Spread Out Brag peak (E-SOBP) assuming that the reference radiation is a radiation field in the middle of the SOBP.
EOBP caused complex DNA…change to …. Proton field at E-SOBP….
Page 2
…..Protons are heavily charged particles…….change to …….Protons are charged particles ….(proton is not heavily charged Z=1, M=1 )
240kbp x-ray photon radiation…change to …..240kVp x-ray photon radiation
Page 3
….considerably affects the generation in density of secondary electrons, that are crucial in terms of energy and quantities which regulate damage at the molecular level…….demand rephrasing …..
…..considerably affects the track structure of protons, that is crucial in terms of DNA damage.
Further, at the distal region of the Bragg Peak, the proton effectiveness grows, leading to more concentrated distributions of dose around each ion path, thus contributing to the induction of more clustered damage and RBE…change to ….Further, at distal region of the Bragg Peak, the proton RBE is growing due to more dense track structure (increasing LET) that is leading to clustered damage.
in the middle and end of the Bragg Peak…change to……in the middle and end of the SOBP…
Since high LET follows “the theory of dual radiation action”;…….it is very wrong , LET is nothing doing with “the theory of dual radiation action”;……. Just say we applied LQM to analyse survival fraction (SF) and derived parameters alpha and beta of LQM from in vitro SF cells experiment.
The experimental groups were no treatment, proton radiation treatment at the M-SOBP and radiation at E-SOBP, where number of sample n=18. Not clear what do you want to say , rephrase please.
The experimental groups were no treatment……do you mean the control group of cells ?
Page 6
However, the highest…..???? not finished sentence
“The RBE was calculated at 10% surviving fraction (SF) and with 2Gy E-SOBP”……… It is not clear , are you talking about two different RBEs: RBE-10 and RBEd ? If you are talking about RBE-10 it is not relevant to D=2Gy at E-SOBP. If you are talking about RBEd it should be related to 2Gy in M-SOBP. You should present a formula for RBE related to 2Gy , it will be important for Fig 3 understanding.
It is not clear what RBE is presented on a Fig 3 and how it was obtained (see above). You are using M-SOBP as a reference radiation and then compare with a single point at E-SOBP (it was not mentioned at which point at the end of SOBP you are deriving RBE. Is it R-90 or R-80? LET is changing dramatically at the distal part of the SOBP. ).Could you please mention max energy of the proton beam and depth at which cells were placed. What is the water equivalent thickness of the cell holder?
I guess that Fig 3 was obtained from SFs presented at Fig 2 as ration of SFs vs dose or ratio of doses for variable dose at E-SOBP, could you please provide explanation.. Why in a Fig 3 B only two curves presented? Whats is error bars for these experimental data?.
Fig 2 is presented in a linear scale that is not acceptable and do not allow to see difference in SF of MOBP (M-SOBP) and EOBP (E-SOBP) for 1% SF. Please rescale using a log scale on a vertical axis.
“A linear quadratic (LQ) fit for cell survival was generated to calculate the RBE be-tween M-SOBP and E-SOBP. Thus, the data delineates RBE was higher at E-SOBP as com-pared to M-SOBP, indicating more cellular damage than at E-SOBP. As a standard practice 2Gy radiation is commonly given in the clinic and at this 2Gy level, the RBE at E-SOBP was also higher at 1.67 for U-CH2 and 1.55 for Mug-chor1 (NEEDS ERRORS). This sug-gests there is a significantly higher RBE at the E-SOBP than the M-SOBP in chordoma cell lines killing” …… M-SOBP is a reference radiation , i.e. possible to say about RBE of radiation field at E-SOBP only.
“2Gy radiation is commonly given in the clinic and according to 2Gy the RBE at E-SOBP is also higher at 1.67 for U-CH2 and 1.55 for Mug-chor1 (Table 2).”…Table 2 did not present these data. Possibly you mean Fig 3. Please add these data to the Table 1 and 2
Please introduce errors for all found parameters in a Table 1 and 2.
Author Response
Dear sir,
Thank you for your insightful comments and we have tried our best address all comments. We have changed figures and tables and also reformatted the manuscript. We are sure this amnuscript will attract wide scientific audiance.

Reviewer 3 Report
In this study, using clonogenic assay, the authors evaluated the survival of chordoma U-CH2 and Mug-Chor1 cells after been exposed to proton radiation at the middle (M-) or at the end (E-) of the Spread-Out Bragg Peak (SOBP). Gene expressions in U-CH2 cells post different treatments were analyzed. They found that (i) less cell survival of both cell lines at the E-SOBP versus at the M-SOBP; (ii) in U-CH2 cells, post E-SOBP, the expression of CASP8, CDKN1C, Ras Suppressor Protein 1 genes were upregulated; the expression of RAD51B gene was down regulated; the OX40 L pathway was down regulated. They claimed that (a) E-SOBP could be clinically more relevant than M-SOBP; (b) E-SOBP causes more chordoma cell death due to the insufficiently activated DNA recombinational repair; (c) agonist anti-OX40 antibodies could be used to block OX40 receptor from binding its ligand and stop chordoma cell growth.
This type of original researches in the area of proton therapy are important. The findings derived from these studies will provide evidences of potential clinically relevant optimization of proton treatment plans and identify potential targets to improve the radiotherapy outcomes in patients with malignancies. However, in the current study, the following numerous deficiencies generated confusion and need major revisions:
Major revisions:
- The writing of this manuscript (including the title, the abstract, the descriptions of the results, and the discussion) is confusion and with multiple repeating or mistakes, scientific publication editing service is necessary.
- On page 3, in line 13, only U-CH2 cells have been tested.
- The description of radiation settings for E-SOBP is needed.
- Please list the Mev of the protons, and please clarify if the radiation doses were physical doses.
- Please list how many independent repeating experiments have been done.
- The validation of Microarray results is necessary. Please add these information.
- Please add the information of cell collection time point for Microarray experiment because gene expression could be time point dependent.
- Please define the RBE calculation method at 2-Gy for the E-SOBP.
- Please quote Table2 in the text.
- Do you have Lower CI for Figure 3 B? If yes, please add it.
- Inaccurately using the word “relative biological effectiveness (RBE)” has generated confusion and influence the accuracy of descriptions through the whole manuscript. In some places, the “RBE” can be used, but in other places, the “biological effectiveness” should be used. Please make necessary changes.
- Please clarify if the comparison of gene expression was between M-SOBP and E-SOBP. Have you compared the treatment group with the no treatment control group?
- Please clarify which treatment modality will benefit more from the agonist anti-OX40 antibodies.
- Gene expression could be different from cell line to cell line. Findings from one single cell line are not necessary to represent a general concept. Please add Microarray analysis of Mug-Chor1 cell line to identify general changes post M-SOBP and E-SOBP.
- The Discussion and Conclusion sections are overstated in numerous places. Please re-write these sections to only include findings directly observed from the current study.
- Please add a paragraph to describe the limitations of this study.
Minor revisions:
There are numerous typos and errors in the manuscript. Please make corrections accordingly.
- On page 1, Abstract: missing “(1)” in front of “Background”.
- On page 2, in line 3, change “bream” to “beam”.
- On page 2, in the 2nd paragraph, x-ray is not a charged particle.
- On page 6, in the 1st paragraph, change “2.1” to “3.1”.
- On page 7, “Figure 3. Radiobiological Effect (RBE)”, please delete “(RBE)”.
Author Response
Dear sir,
Thank you very much for reviewing our manuscript and we have duly addressed all comments. We have incorporated new figs and tables and added new section in discussion. We are sure this manuscript will generate wide interest among scientic community.
Sincerely,
Hem D Shukla

Round 2
Reviewer 2 Report
Please see attachment.

Author Response
Dear sir,
We kindly agree with reviewer comment and we have corrected the equation to calculate RBE as follows, and we have also corrected in the manuscript.
- We have indeed used a formulation of RBE as the ratio of SF at a fixed dose. We amended the final sentence of section 2.7 to read:
- “Using the fitted cell survival curves for the E-SOBP and M-SOBP and , the relative biological effectiveness was calculated: RBE(D) = SFE-SOBP(D)/SFM-SOBP(D), where D is the radiation dose.”
- We apologize and have removed “hyperthermia” as this was an error.
